# An Unbiased Look at Datasets for Visuo-Motor Pre-Training

**Sudeep Dasari**[*]      **Mohan Kumar Srirama**      **Unnat Jain**[†]      **Abhinav Gupta**[†]
CMU                          CMU                        FAIR at Meta               CMU

https://data4robotics.github.io

**Abstract:** Visual representation learning hold great promise for robotics, but is severely hampered by the scarcity and homogeneity of robotics datasets. Recent works address this problem by pre-training visual representations on large-scale but out-of-domain data (e.g., videos of egocentric interactions) and then *transferring* them to target robotics tasks. While the field is heavily focused on developing better pre-training algorithms, we find that dataset choice is just as important to this paradigm's success. After all, the representation can only learn the structures or priors present in the pre-training dataset. To this end, we flip the focus on algorithms, and instead conduct a *dataset centric* analysis of robotic pre-training. Our findings call into question some common wisdom in the field. We observe that traditional vision datasets (like ImageNet, Kinetics and 100 Days of Hands) are surprisingly competitive options for visuo-motor representation learning, and that the pre-training dataset's image distribution matters more than its size. Finally, we show that common simulation benchmarks are not a reliable proxy for real world performance and that simple regularization strategies can dramatically improve real world policy learning.

**Keywords:** Visual Representation Learning, Datasets, Robotic Manipulation

## 1 Introduction

Consider a robot that must perform a manipulation task in an unstructured environment: e.g., toasting a bread slice. To accomplish this, the robot must locate the target objects (bread, toaster, etc.) in the scene and reason about their physical properties (e.g., Center-of-Mass, etc.), using RGB camera inputs. However, the real world has innumerable objects, lightning conditions, and environments that a robot may run into. This incredible range of scenarios makes hand-engineering a vision pipeline impossible. Thankfully, the computer vision and representation learning communities have highlighted a successful paradigm to overcome this challenge: learn end-to-end neural representations *directly from data* [2, 3], which can then be used for downstream vision tasks. We seek to do the same for policy learning.

But what data should these representations be trained on? In an ideal world, we would leverage task-specific robotic data (i.e., trajectories) to jointly learn a visual representation and a controller, using end-to-end reinforcement or imitation learning [4, 5, 6, 7]. Unfortunately, learning visual representations in conjunction with action policies is frequently intractable [8, 9] or requires a large amount of data, that may be too expensive to collect on real hardware. Furthermore, the homogeneity of robotics data

## What Data Should Robots Be Pre-Trained On?

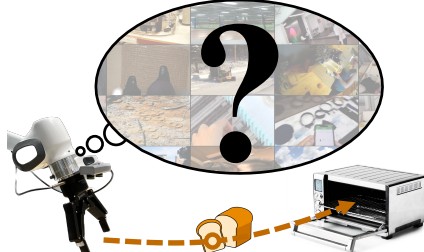

Figure 1: Due to the scarcity of diverse, large-scale robotic data, visuo-motor representations – which are necessary to solve tasks (e.g., put bread in toaster) from visual inputs – must be learned from external datasets [1]. But which datasets contain the best priors for robotics? Surprisingly, we find that simply pre-training on standard vision datasets (e.g., ImageNet) can outperform SOTA baseline representations from the robot learning community, despite using roughly 5x less data.

---

[*]First/Corresponding Author: *sdasari@cs.cmu.edu*
[†]Denotes equal advising

7th Conference on Robot Learning (CoRL 2023), Atlanta, USA.

(collected in single lab) hinders generalization to novel scenarios, which is the motivation for learning in the first place! To overcome this, the field has trended towards *pre-training* visual representations on large-scale, unlabeled vision datasets, using self-supervised learning algorithms – e.g., Masked Auto-Encoders [10] (MAEs), contrastive learning [11, 12, 13], etc. These pre-trained representations decouple policy learning from perception, allowing us to learn behaviors with far less robotic data [1, 14, 15, 16, 17].

The key insight is that self-supervised visual pre-training can learn useful priors from *out-of-domain* data that will be useful for robotics. Recent work in robot manipulation [14, 1, 18, 15] has investigated different neural architectures and algorithms for learning these priors. However, there is one important commonality – all these methods train (primarily) on the same dataset, Ego4D [19]. Indeed, this seems like an intuitive choice, because: (1) Ego4D contains first-person camera views, which are analogous to the robot's camera; (2) Ego4D focuses on human-object interaction – i.e., it is aligned with the downstream manipulation task; and (3) the dataset offers thousands of hours of video frames to train on. But is this *intuitive bias* empirically tested?

In this paper, we empirically investigate these research questions from the perspective of robotic manipulation tasks. Specifically, we pre-train a total of 15 representations on various datasets using MAEs [10], a state-of-the-art (SOTA) self-supervised learning algorithm. We then fine-tune each of these representations to solve various manipulation tasks in simulated and real settings via Behavior Cloning (w/ $\leq 50$ demonstrations). Our experiments reveal that many intuitive biases and **common assumptions in our field need to be revisited**. Surprisingly, we find that standard image datasets based on curated internet data (e.g., ImageNet [2], Kinetics [20], 100 Days of Hands [21]) can learn stronger visuo-motor representations than egocentric human interaction data (of Ego4D)! In fact, pre-training on the ImageNet compares favorably against SOTA (visuo-motor) baseline representations, which were trained on far more data (e.g. MVP [22] was trained on 2M+ frames) using the exact same algorithm and hyperparameters. This leads us to an importance insight – the pre-training image distribution is far more crucial for effective representation learning than naively increasing the number of images to train on. Building on this, we investigate various schemes for scaling pre-training dataset size while creating a broader image distribution. Our best model improves performance by $30\%$ (v.s. SOTA baselines [15, 22]) on real world robotics tasks and is the direct result of this search. Finally, we show how simple implementation details (like using dropout [23] during evaluation) can have a significant impact on policy performance, and how these trends are poorly captured in simulation studies. Our project code and models are released publicly, and we encourage the reader to view our website for added context[3].

## 2 Related Works

**Learning Actionable Representations** The robotics field has long focused on learning actionable representations, which focus on task relevant details and are maximally predictive of the actions the robot should take. These representations can be learned end-to-end as part of policy learning, using data collected by expert demonstrations (e.g., Imitation Learning [24]) or the robot itself (e.g., Reinforcement Learning [4]). This paradigm has been successfully applied to a wide range of tasks like in-the-wild grasping [25], bin-picking [26, 27, 28], insertion [5], pick-place [29], and even self-driving [30, 31, 8]. Prior work also added tertiary optimization objectives (e.g. observation reconstruction [32], inverse modeling [33], dynamics modeling [34], etc.) on top of policy learning, in order to make representation learning more efficient. However, all of these techniques share the same flaw: they require a wealth of task-specific robotic data for learning representations.

**Self-Supervised Visual Pre-Training** Thus, the robotics community has trended towards *pre-training* representations on out-of-domain, vision datasets, (which are both larger and more diverse) and transferring them to robotics tasks. Prior works [1, 18, 22, 14, 15] all seem to follow a common formula: representations are trained using SOTA self-supervised vision algorithms (e.g., contrastive learning [11, 13, 12], masked image modeling [10, 35], etc.) on frames (primarily) sampled from the Ego4D dataset [19]. These representations are then evaluated mostly in sim [36], using a common policy learning framework [1, 15]. These choices may seem reasonable (see Sec. 1), but there is surprisingly little evidence backing them. *Importantly, R3M [1] and MVP [14] compared only with supervised ImageNet representations but not apples-to-apples with self-supervised ImageNet representations [10].* Our investigation fills in these critical gaps. We find that representations

---

[3]https://data4robotics.github.io

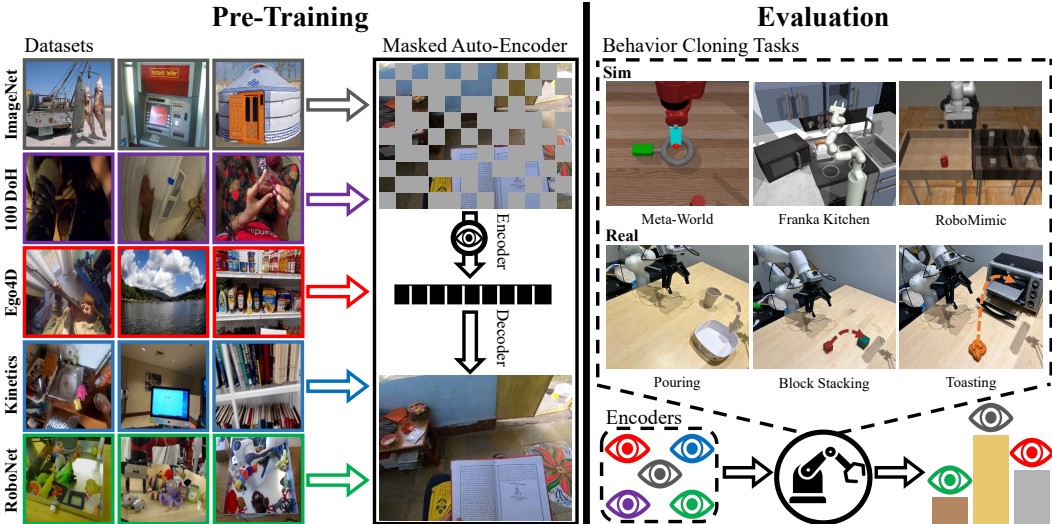

Figure 2: Our investigation considers 5 standard datasets from both the computer vision and robotics: ImageNet [2], 100 Days of Hands [21] (DoH), Ego4D [19], Kinetics [20], and RoboNet [37] (left). For each dataset, we pre-train a visual representation on it using the Masked Auto-Encoders (MAE) algorithm [10]. This masked image modeling method works by randomly masking patches in the image, and training an encoder/decoder to reconstruct them (center). Once pre-training is concluded, we fine-tune the representation to various robotics tasks, both in sim and in the real world (right).

learned on standard image datasets (like ImageNet) are surprisingly applicable to the robotics space, and that common evaluation/experimental techniques can give a misleading sense of progress.

## 3 Experimental Methods

Our investigation follows a simple formula (see Fig. 2). Step 1: We pre-train visual representations on various datasets using the same self-supervised algorithm (masked image modeling). Step 2: We fine-tune each representation for downstream manipulation tasks in both simulated and real (via behavior cloning). For evaluation of representation quality, we rate based on performance on downstream tasks, with emphasis on performance in the real world.

**Visual Pre-Training** This project requires a scalable representation learning algorithm that can seamlessly operate on heterogeneous data sources, with the highest possible performance. We chose to use Masked Auto-Encoders (MAEs), a self-supervised representation learning algorithm with SOTA performance on various vision [10, 35, 38, 39, 40, 41, 42], multimodal [43, 44], and robotics tasks [15, 22]. The MAE encoder ($E$) is a Vision Transformer (ViT) network [45] that produces an embedding vector to represent an input image $I$: i.e. $E(I) \in \mathcal{R}^{768}$. During training $E$ is tasked to represent $I$, using only 25% *random patches* sampled from the image. Then a decoder network ($D$) attempts to reconstruct $I$ in its entirety (see Fig. 1, middle). Both $E$ and $D$ are trained end-to-end, minimizing the MSE reconstruction objective: $||D(E(I)) - I||_2$. During training, the visual encoder learns to reason spatially: i.e., it learns how patches relate to each other, and how they can come together to form the final image. Thus, $E$ learns a highly efficient image descriptor that can be transferred to downstream tasks without any algorithmic changes (e.g. no masking needed during transfer). The MAE hyperparameters are described in Appendix A. Note that they are directly copied from the original MAE work by He *et al.* [10] and shared by prior works in robot learning [15, 22].

**Fine-Tuning w/ Behavior Cloning** Pre-trained visual representations are fine-tuned to solve downstream tasks of robotic manipulation. To this end, we adopt the paradigm of *Learning from Demonstration (LfD)* [46, 47, 24, 48, 49, 50]. Our goal is to learn a policy $\pi$ that uses the given observation $o_t$ to predict an optimal action distribution for the task: $a_t \sim \pi(\cdot|o_t)$. Note that the actions $a_t$ are commands sent to the robot controller, while the observations consist of the current image and robot joint information: $o_t = [i_t, j_t]$. The policy $\pi$ must be learned given a set of expert demonstrations ($\mathcal{D} = \{\tau_1, \ldots, \tau_n\}$), where each demonstration $\tau_i = [(o_0, a_0), \ldots, (o_T, a_T)]$ is a trajectory with optimal observation-action tuples (i.e. collected by a proficient agent).

$\pi$ is parameterized using a 2-layer network ($p$), built atop the pre-trained encoder $E$. The forward pass works as follows: first, the observation image is encoded $E(i_t)$; then $j_t$ is concatenated to the encoding and passed to the policy network – $p(E(i_t), j_t)$. $p$ predicts a policy distribution, which in our case is a Gaussian Mixture Model [51, 52]. During test time, actions are sampled from this distribution and then executed on the robot. The entire policy network (both $p$ and $E$) is fine-tuned end-to-end (using Adam [53] for $50K$ iterations) to maximize the log probability of actions from the expert demonstrations: $\min_\pi -log(\pi(a_t|E(i_t), j_t))$. This procedure was designed to closely match prior work with two important modifications: we apply dropout to the policy network $p$, and we apply data augmentation to $i_t$ before passing it to the encoder. Both of these deviations are validated in our experiments (see Sec. 5). Please refer to Appendix B for the exact hyperparameters.

**Evaluation Procedure**  To evaluate a representation, we apply the above fine-tuning stack separately on 13 sim tasks and 3 real tasks. Each policy's final checkpoint is evaluated on $N$ test rollouts for every task, with novel initializations (e.g., test objects, new initial positions, etc.). All evaluation hyperparameters (e.g., demonstration set, number of test-time rollouts, initial positions, test objects, BC hyperparameters, etc.) are kept fixed within a task. This allows for maximally fair evaluation.

**Simulation Tasks:** Our simulated tasks set spans a set of 3 MuJoCo [36] environments – Meta-World [54], RoboSuite [51], Franka Kitchen [55] – that are frequently used by the robot learning community, and the exact setups (e.g., task rewards/success criteria, camera positioning, object sets, demonstration trajectories, etc.) were directly taken from prior work [15, 1, 51] (fully documented in Appendix C). As a result, our simulated results should be very accessible to the community.

**Real World Tasks:** While simulation is a useful tool, there is a significant sim2real gap in manipulation. Thus, we designed 3 distinct tasks for real world validation on a Franka Panda Robot (visualized in Fig. 2).
(1) *Block Stacking* requires the robot to pick up the red block and place it on the green block. This is the simplest of three tasks as the robot only has to adapt to new object configurations during test time. However, the robot still needs to precisely localize and grasp the (small) red block.
(2) *Pouring* requires the robot to lift the cup and pour almonds in the target bowl. At test time, the cup and target bowls are both novel objects (unseen during training), and are placed in random positions, requiring the robot to generalize to new visual inputs.
(3) *Toasting* is our most challenging task, and it requires the robot to pick up the object, place it in the toaster, and then shut the toaster. At test time, we use a novel object and randomize both the object's initial pose and the toaster's initial orientation. Toasting requires the robot to execute a multi-stage manipulation strategy, while also generalizing to new visual scenarios.
Each of the three tasks use a **shared action space:** Cartesian velocity control; and a **shared observation space:** proprioceptive inputs and 3rd person camera view (visualized in Fig. 3, left). We collect $n = 50$ tele-op demonstrations per task. Please refer to Appendix C for all other task hyperparameters and our website for task videos: https://data4robotics.github.io.

## 4  Probing Dataset Biases

In our empirical study, we evaluate 5 widely-used datasets as pre-training candidates (see Fig. 2, left): ImageNet [2], Ego4D [19], 100 Days of Hands [21] (DoH), Kinetics [20], and RoboNet [37] (see Sec. 4.1 for descriptions). We apply our experimental methodology (from Sec. 3) on various sub-samplings/combinations of these datasets. First, we conduct single dataset pre-training: i.e., we evaluate a dataset's performance in isolation to empirically determine which is most suited for our diverse downstream manipulation tasks (Sec. 4.1). In our second suite of experiments, we analyze how well various combinations of the data perform (Sec. 4.2). Finally, we investigate dataset scaling for pre-training, and find that the pre-training image distributions matter most (Sec. 4.3).

### 4.1  Comparing Datasets Apples-to-Apples

Before diving into the details, let's take a step back and add context about the datasets we probe.
(1) **ImageNet** (ImageNet-1K) contains 1000 train images for each of its 1000 classes. ImageNet is a popular and classical computer vision dataset, i.e., curated carefully from internet images. The broad image distribution may result in more expressive representations (as observed in purely visual tasks like classification). However, ImageNet is focus on centered, single-object, and high-quality internet images. As a result of this domain gap, many believe that ImageNet is an ill fit for robotics.

## Robot Observations vs Pre-Train Image Distributions

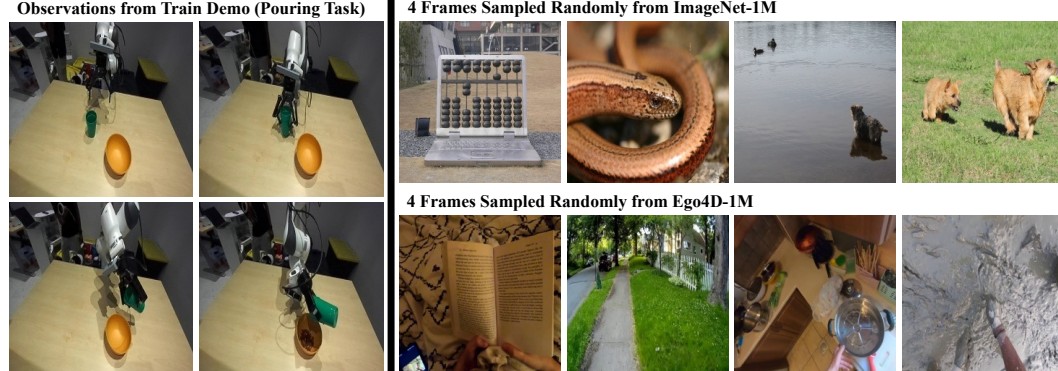

Figure 3: Observations from our pouring task (left) are compared against random pre-training images from ImageNet-1M/Ego4D-1M (right). Note that all the pre-train images are very different from the evaluation task. Nonetheless, the curated, single-object images from ImageNet-1M yield stronger visuo-motor representations than the Ego4D-1M frames do (see Table 1).

| | Task | Single Dataset Models (1M Images) | | | | | Baselines | | |
| | | ImageNet | Ego4D | Kinetics | 100 DoH | RoboNet | Scratch | VC-1 [15] | MVP [22] |
|---|---|---|---|---|---|---|---|---|---|
| **Sim** | RoboSuite [51] | 62% | 61% | 65% | 52% | 58% | 2% | 63% | 51% |
| | MetaWorld [54] | 90% | 93% | 87% | 86% | 74% | 72% | 70% | 83% |
| | Franka Kitchen [55] | 63% | 68% | 55% | 62% | 62% | 40% | 61% | 61% |
| | **Average (Sim)** | 72% | 74% | 69% | 67% | 65% | 38% | 65% | 65% |
| **Real** | Block Stacking | 68% ± 9.5% | 64% ± 9.5% | 72% ± 9.8% | 55% ± 9.2% | 76% ± 8.7% | 0% ± 0% | 60% ± 10% | 52% ± 10% |
| | Pouring | 44% ± 12% | 19% ± 9.0% | 50% ± 9.0% | 19% ± 12% | 13% ± 7.6% | 0% ± 0% | 25% ± 10% | 13% ± 7.6% |
| | Toasting | 10% ± 16% | 0% ± 0% | 10% ± 16% | 40% ± 17% | 0% ± 0% | 0% ± 0% | 10% ± 10% | 10% ± 10% |
| | **Average (Real)** | **41**% ± 7.1% | 28% ± 6.9% | **44**% ± 7.1% | **38**% ± 6.9% | 30% ± 7.0% | 0% ± 0% | 33% ± 6.9% | 25% ± 6.6% |

Table 1: **Comparing Datasets Apples-to-Apples.** We compare pretrained representations, learned on 1M images from five datasets. We report success rates after finetuning representations with BC, and the real world evaluations also include standard error (i.e., Success% ± Std. Err.%). For additional context, we benchmark SOTA baselines [15, 22] and a "Scratch" representation with no pretaining. We find that visual representations learned on standard vision datasets with internet images and curation (e.g., ImageNet) provide surprisingly strong performance in the real world.

(2) **Ego4D** is a modern, ego-centric, and in-the-wild video dataset, with 3.6K hrs of video collected by humans performing daily tasks. It is conjectured that Ego4D is well suited for robot learning as it contains realistic images that a robot may observe in real-world environments. However, frames collected within the same video tend to look similar to each other, and the lack of curation mean that some object classes (e.g. blocks, cups, etc.) rarely appear.

(3) **DoH** is focussed on human hands and contains curated YouTube videos of people manipulating various household objects (e.g. in cooking video). The curation ensures that action classes are balanced and that videos look distinct from each other. Furthermore, the focus on manipulation may help the representations pick useful cues (e.g. where objects can be grasped). However, YouTube videos look quite different from robot's visual observations, so its an open question if priors learned from DoH would benefit downstream tasks in robotics.

(4) **Kinetics**(-700) is similar to DoH in that it's curated from YouTube, but its videos contain a much wider distribution of human actions (e.g. with objects and/or other humans) instead of the focus of DoH on hands and manipulation.

(5) **RoboNet** contains 13M+ image observations of robots randomly interacting with objects placed in a bin in front of them. RoboNet could be invaluable for pre-training, since its images are highly domain-specific for our use case. But robot data is collected in sterile lab setting, which could cause the representations to overfit to only those specific settings.

It is clear that these five datasets have complex trade-offs that may affect their usability for robotics. However, none of them clearly match the robot observation space (see Fig. 3). The only way to settle the question is to undertake an unbiased and empirical study, comparing them apples-to-apples. Thus, we apply our evaluation methodology to 1 Million frames sampled randomly from every dataset. This is easy to do in ImageNet, since it has 1M balanced train images. But for the video

| | Task | Soup-1M | Soup 2M | Soup-1M + 1M Extra Frames (2M Total) | | | | |
|---|---|---|---|---|---|---|---|---|
| | | | | ImageNet | Ego4D | Kinetics | 100 DoH | RoboNet |
| Sim | RoboSuite [51] | 64 | 64 | 52 | 53 | 59 | 67 | 58 |
| | MetaWorld [54] | 87 | 89 | 92 | 86 | 87 | 92 | 88 |
| | Franka Kitchen [55] | 66 | 67 | 56 | 61 | 64 | 62 | 60 |
| | **Average (Sim)** | 72 | 73 | 67 | 67 | 70 | 74 | 69 |
| Real | Block Stacking | $76\% \pm 8.7\%$ | $44\% \pm 10\%$ | $72\% \pm 9.2\%$ | $60\% \pm 10\%$ | $76\% \pm 8.7\%$ | $\mathbf{92}\% \pm 5.5\%$ | $76\% \pm 8.7\%$ |
| | Pouring | $38\% \pm 13\%$ | $38 \pm 13\%$ | $32\% \pm 12\%$ | $38\% \pm 13\%$ | $32\% \pm 12\%$ | $\mathbf{38}\% \pm 13\%$ | $32\% \pm 12\%$ |
| | Toasting | $10\% \pm 10\%$ | $40\% \pm 16\%$ | $0\% \pm 0\%$ | $10\% \pm 10\%$ | $22\% \pm 13\%$ | $\mathbf{50}\% \pm 17\%$ | $0\% \pm 0\%$ |
| | **Average (Real)** | $41\% \pm 7.1\%$ | $41\% \pm 7.0\%$ | $35\% \pm 7.1\%$ | $36\% \pm 7.1\%$ | $43\% \pm 7.1\%$ | $\mathbf{60}\% \pm 6.7\%$ | $36\% \pm 7.1\%$ |

Table 2: **Marginal Value of Each Dataset.** Soup-{1M,2M} models are trained {1M,2M} images with {200K,400K} images from each of the five target datasets. The models on the right are trained with the Soup 1M images and an additional 1M frames from the target dataset. We find that image distribution matters more than the number of images trained on: Soup-2M does not improve on Soup-1M, but Soup-1M + 1M DoH does. Results are reported as success rates for each task, and the real world evaluations also report standard error (i.e., Success% ± Std. Err.%).

datasets (Ego4D/Kinetics/DoH), we first processed them into frames (sub-sampled at 3FPS) and then randomly select 1M images from the whole set. For RoboNet, we followed a similar procedure as used for the video datasets, but randomly sampled 1M image observations instead. Visual pre-training on each of these results in 5 representations that we evaluate on our task suite (via BC). For additional context, we also evaluate a '*Scratch*' model with no pre-training, i.e., randomly initialized weights before BC. We also evaluate pre-trained weights downloaded from two SOTA baselines (VC-1 [15], MVP [22]). Note that both MVP and VC-1 share our vision transformer architecture and pre-training recipe of masked image modeling, but were trained on significantly more # of frames (2.5M+), sampled primarily from Ego4D. The results are presented in Table 1.

Our first observation is that performance trends from popular simulation benchmarks do not transfer to the real world at all (see Sec. 5 for more). Thus, we focus the rest of our analysis on the real world trends, since that is the primary focus of this work. The real robot experiments reveal that **ImageNet/Kinetics/DoH representations all perform better than those trained on RoboNet/Ego4D** (roughly $40\%$ v.s. $30\%$ success rate). Critically, this result goes beyond just MAE pre-training. As we show in Appendix D, our finding that ImageNet/Kinetics/DoH performs best *also holds with contrastive pre-training* [13]! This is surprising and important since both Ego4D and RoboNet seem like better matches to the downstream tasks (e.g., RoboNet entirely contains images of robot interactions) and as more works in the research community implicitly assume/expect Ego4D to do better. Note that ImageNet/Kinetics/DoH were all sampled and curated from the internet (using YouTube/image search), so they contain cleaner images with a much greater range of content (e.g., 1000 classes [2] vs 4 robot labs [37]). These unbiased, empirical results strongly suggest that the **pre-training image *distribution* is far more important than the images' *content*.**

## 4.2 Combining Data from Different Sources

Another surprising result from Table 1 is that the baselines representations perform worse than the ImageNet/Kinetics/DoH representations, despite being trained on significantly more images. For example, VC-1 pre-trains on ImageNet alongside 2.5M+ images from Ego4D, while using the exact same pre-training strategy that we do. A possible explanation for this discrepancy is that VC-1's representation is functionally very similar to our only-Ego4D ablation, since the majority of its pre-training frames come from Ego4D. Consequently, each batch the encoder sees during pre-training primarily consists of Ego4D frames. The key insight here is that *distribution* of the pre-training set matters more than the sheer number of frames trained on. We experimentally test this hypothesis, and find that VC-1's representation performs only marginally better than Ego4D's ($33\%$ vs $28\%$).

The natural next question is, "*How does one optimally combine datasets for visuo-motor pre-training?*" A simple idea is to proportionally mix the datasets so that the model is pre-trained on an equal number of frames from each dataset. Particularly, we create a "Soup-1M" containing $200K$ images randomly sampled from each of the 5 datasets. We then evaluate this model on our test suite (see Table 2, left). Note that the Soup-1M model performs about the same as the ImageNet/Kinetics/DoH models ($41\%$), even though it was trained on a significant amount of Ego4D/RoboNet frames (recall, these performed lowest in Sec. 4.1). **This suggests that scaling to multiple datasets can increase robustness, so long as the datasets are kept carefully balanced during pre-training.**

### 4.3 Analyzing the Marginal Value of Each Dataset

Soup-1M provides a sensible first step for combining datasets for visuo-motor pre-training – keeping training set size to 1M using equal proportions of data sources. This leads to the natural next question: "how can we effectively scale dataset size to improve performance?" To answer this question, we'll also need to understand the marginal value of adding additional data to the soup. To answer this, we undertake another empirical study. Particularly, we obtain visual representations on pre-training sets containing the aforementioned Soup-1M along with 1M images from each of the five subject datasets (e.g. Soup-1M + 1M ImageNet frames). In effect, this both scales the size of pre-training dataset, while shifting the train distribution towards that of the subject dataset. For fair comparison, we also train a Soup-2M model (identical to Soup-1M but with $400K$ images per dataset) that tests a naive scaling of the Soup-1M model. All six models are evaluated and results are present in Table 2.

As reported in Table 2 (left), we find that Soup-2M model performs marginally better in simulation than Soup-1M, and performs exactly the same (on average) in the real world. That is, **data scaling is more nuanced than naively increasing the number of frames**. In contrast, the strongest model, Soup-1M + 1M DoH, is able to perform $20\%$ better than Soup-1M (and $30\%$ better than the strongest baseline) on the real world tasks (bold results in Table 2)! Finally, the Soup-1M + 1M {Ego4D/ImageNet/RoboNet} models perform slightly *worse* than Soup-1M, whereas the Soup-1M + 1M Kinetics model performs slightly better. These results are mostly in line with our expectations from Sec. 4.1 (e.g. adding more RoboNet data reduces performance, while adding more Kinetics/-DoH increases performance).

## 5 Ablating our Experimental Setup

This section presents some insights from our real-world experiments. We find: (1) that old-school dropout regularization is highly effective; and (2) sim *evaluation* does not transfer to real world.

**Regularizing Policies w/ Dropout**   Early in our physical robot evaluations, we noticed that the policies often produced jerky motions that could damage the robot and its environment. Thus, we searched for a simple fix that could improve robustness in the real world. We found that adding dropout [23] to the policy network (w/ $p = 0.2$) significantly improved the robot's qualitative behavior: the commanded motions became smoother, with improved generalization to new scenarios. We quantify this with ablations on the *Block Stacking* task, i.e., fine-tuning the five 1M models (see Sec. 4.1) with and without dropout. Note how **adding dropout to visuo-motor policies almost consistently improves policy performance on the physical robot** (see Fig. 4, orange bars). However, the opposite effect is observed in simulation. This indicates that adopting sim benchmarks as a (fast) proxy to make policy design choices may warrant caution and a healthy doze of skepticism.

We further test the regularization effect of dropout in this setting using a new task: *Block Stacking Robust*. In this task, a human adversary pushes the cube out of the robot's gripper in the middle of the episode (i.e. right before the robot grasps). This forces the robot to dynamically replan its actions, and adapt to a scenario it never saw during fine-tuning with BC (in the demonstrations). We find that the success rate on *Block Stacking Robust* (average across all models) increases to **24**% from $10\%$ thanks to this regularization.

**Analyzing Sim-to-Real Transfer**   On one hand the sim2real gap in manipulation is well known and on the other it's still very common practice in prior work [1, 15, 22, 18, 56] to draw inferences about pre-trained representations using simulated benchmarks (e.g. CortexBench [15], Franka Kitchen [55], Isaac Gym [22], etc.) In several unbiased experiments we undertook, we have found that trends in simulation are almost entirely disconnected from their real world performance. *First*, the simulation suite predicts that Ego4D is the best representation for robotics, but the real world results consistently disagree with that assessment (see Table 1). *Second*, key implementation details in the real world (like Dropout) can actually hurt performance in simulation (see how Dropout hurts sim performance in Fig. 4). To objectively investigate this, we plot the sim performance vs real performance for all our models (trained on dataset configurations detailed in Sec. 4) in Fig. 5. We find that **sim and real performance are almost entirely uncorrelated (a very low $R^2$ = 32%).** Even if we were to 'cherry-pick' the two most similar sim/real tasks (RoboMimic's block-lift [51] vs our stacking task) the correlation is still very low: $R^2 = 34\%$.

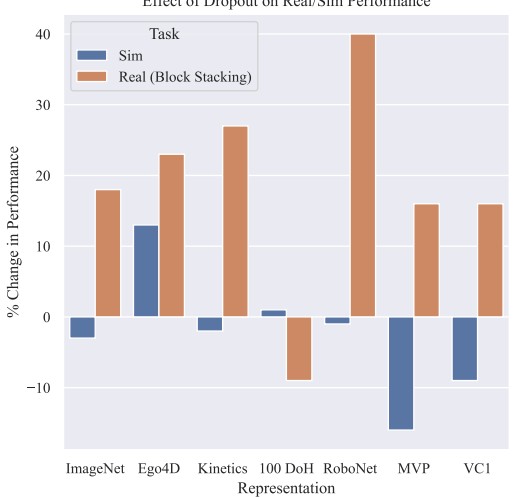

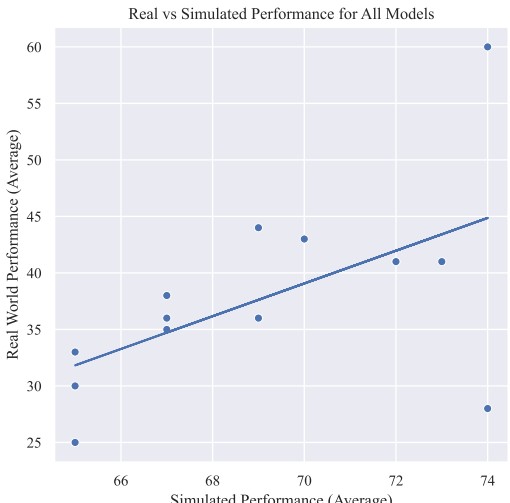

Figure 4: **Effect of adding dropout [23] in sim *vs.* real** (block stacking tasks). Dropout frequently harms performance in simulation (blue) but consistently improves real world (orange) success rate. Positive values on Y axis indicate improvement by adding dropout and vice-versa.

Figure 5: **Sim vs. real performance across pretraing datasets.** We plot average model performance, in sim and real, for all the models tested in our study. Note how the sim scores are only weakly predictive of real world performance ($R^2 = 32\%$).

## 6 Discussion

In this paper, we investigated how dataset biases and implementation choices can affect visual pre-training for robotics. With the modus operandi of masked image modeling [10], our experiments analyzed pre-trained visuo-motor representations trained on 15 different data distributions, sourced from 5 common computer vision and robotics datasets. These models were evaluated on standard sim environments, alongside 3 unique and challenging real world tasks (each with 50+ robot evaluations, for rigor). We find that traditional computer vision datasets (e.g. ImageNet) provide surprisingly strong performance for robotic pre-training. In fact, our simple ImageNet representations outperform both Ego4D representations and representative baselines in the field. The key insight is that the image distribution matters much more than the sheer number of images during pre-training. Guided by this insight, we explore various strategies for scaling data distribution, by carefully mixing data from different sources. As part of this investigation, we train a final model (Soup-1M + 1M DoH) that exhibits a 30% improvement over the baselines on real world robotic tasks. Finally, we analyze our experimental methodology, and show how simple regularization techniques (e.g. dropout [23]) can boost real world performance, and conclude that trends in simulation do not correlate to real world deployment. We hope that our unbiased empirical probes and associated findings will inspire others in the field to study how various sources of offline data can transfer to robotics tasks. To enable future efforts, we released all of our pre-trained representations and evaluation code on our website[4].

**Limitations and Future Work** While our experiments were extensive, there are some limitations that should be addressed by future work. First, there is no simple theory or experimental test that can predict if a representation will actually work well on the robot after pre-training. In fact, our experiments show that the easiest evaluation technique, i.e. the proxy of simulation, may give a misleading sense of progress in the real world! Thus, it is vital for our community to find faster ways to evaluate representations, and share reproducible results. One possibility is a standardized cloud robotics benchmark [57, 58] that could greatly reduce the load for researchers. Next, our experiments heavily focused on Behavior Cloning combined with MAE pre-training (though we did explore SimCLR pre-training in Appendix D). Finally, it would be valuable to extend our study in more scenarios (e.g., Reinforcement Learning), and on other robotic tasks, like visual navigation and grasping.

---

[4]https://data4robotics.github.io

**Acknowledgments**

We'd like to recognize our CMU colleagues – Shubham Tulsiani, Jason Y. Zhang, Shikhar Bahl, Russell Mendonca, Yufei Ye, Alex Li, and Mihir Prabhudesai – whose feedback and comments helped strengthen the paper. We are grateful to Xinlei Chen for guidance towards MAE pre-training and self-supervised learning adaptations to the `pycls` codebase [59, 60, 61]. Finally, SD's PhD research is generously supported by the NDSEG fellowship.

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

| Hyperparameter | Value |
|---|---|
| *MAE Pretraining* | |
| optimizer | AdamW [53] |
| base learning rate | 1e-4 |
| weight decay | 0.05 |
| optimizer momentum | $\beta_1, \beta_2 = 0.9, 0.95$ |
| batch size | 4096 |
| learning rate schedule | cosine decay |
| total batches or iterations | 249600 |
| warmup iterations | $1/8 \times$ total iterations |
| augmentation | RandomResizedCrop |
| #GPU | 64 V100 (32 gb) |
| Wall-clock time | $\sim$36 hours |
| *Encoder ViT Architecture* | |
| #layers | 12 |
| #MHSA heads | 12 |
| hidden dim | 768 |
| class token | yes |
| positional encoding | sin cos |
| *Decoder ViT Architecture* | |
| #layers | 8 |
| #MHSA heads | 16 |
| hidden dim | 512 |
| class token used | yes |
| positional encoding | sin cos |

Table 3: Training and architectural hyperparameters for MAE pretraining.

## A  MAE Hyperparameters

We list key hyperparameters for the MAE training loop in Table 3. Note that these parameters were employed directly from original MAE paper [10] and are actually shared by relevant robotics baselines [15, 22]. Consistent with the terminology in [10], the employed learning rate is the base learning rate scaled by (total batch size / 256). For a head-on comparison with prior work [10, 15], we train the ViT for iterations equivalent of 800 epochs over ImageNet dataset. This rigorous benchmarking took # GPUs $\times$ wall clock time $\times$ # data ablations $= 64 \times 1.5 \times 12 = 1152$ GPU days.

## B  BC Hyperparameters

The following section describes the hyperparameters used in our behavior cloning loop. As discussed in Sec. 3, the BC policy begins by taking in the image and passing it through the pre-trained encoder to get a representation, $E(i_t)$. That representation is then concatenated to the joint information to get a policy input, $x_t = [E(i_t), j_t]$. The policy input is fed through a 2-layer mlp network, with a batchnorm preceding the first layer, ReLU activations [3], and hidden dimensions of $[512, 512]$. Additionally, we add dropout [23] to the two mlp layers w/ probability $p = 0.2$ after the ReLU activations. The result of the top layer is then passed to 2 linear layers, that predict the mean ($\mu$), mixing parameters ($\phi$), and standard deviation ($\sigma$) of a Gaussian Mixture Model (GMM) distribution w/ $m$ modes:

$$p(x) = \Sigma_{i=1}^m \phi_i N(x|\mu_i, \sigma_i)$$

The choice of GMM was based on prior work [51, 52] that showed it could dramatically improve performance. After some tuning, we used $m = 5$ on the RoboSuite tasks (note their benchmark [51] used $m = 5$) and the real world tasks, since it worked best. However, for Franka Kitchen and MetaWorld, we found no significant difference. As a result, we used $m = 1$ (i.e. standard Gaussian distribution) for those tasks to maximize comparability with prior benchmarks [15, 55].

The policy was optimized for 50000 iterations using the ADAM optimizer [53], with a learning rate of 0.0001 and a L2 weight decay of 0.0001. In addition, we applied data augmentation (random crops and random blur) to the input image $i_t$, before passing it $E$. This was based on recommendations for best practices from Hansen et. al. [56]. The full code for this setup is open-sourced on our website: https://data4robotics.github.io.

## C   Task Hyperparameters

This section describes the hyperparameters made while setting up both sim and real world tasks. All code (for robot/sim environments and BC training) is open sourced: https://data4robotics.github.io.

**Simulation**   We evaluate on 5 tasks from *Metaworld* [54] (BinPick, ButtonPress, Hammering, Drawer Opening, and Assembly), 5 tasks from *Franka Kitchen* [55] (Knob Turning, Door Opening, Light Switch, Microwave, and Sliding Door), and 3 tasks from *RoboSuite* [62, 51] (Lift, Can, and Square). These environments are frequently used by the robot learning community, and the exact setups (e.g., camera positioning, object sets, demonstration trajectories, etc.) were directly taken from prior work [15, 1, 51]. As a result, our simulated results should be very accessible to the community.

The training demonstrations for these tasks were collected by previous work (CortexBench [15], Relay Policy Learning [55], RoboMimic [51] respectively). We fine-tune on $n = 25$ demos for MetaWorld/Franka Kitchen, and $n = 200$ demos on RoboSuite (again to stay consistent with older papers). Task success is measured by the environments themselves, and we get numbers by estimating success rates empirically using 50 test trajectories. Note that we only evaluate the policy at the end of training (unlike some prior work that evaluated multiple times over the course of training). This was done to ensure the sim evaluation setup matched the real world (i.e. we can't evaluate real policies multiple times during training).

**Real World**   As discussed in Sec. 3, our real world tasks were built using a Franka Panda robot, and we collected 50 demonstrations for each task using a VR tele-op setup. We heavily encourage the reader to get a feel for the training data and tasks by viewing the supplemental video on our website: https://data4robotics.github.io.

The following section expands on our real world task descriptions from Sec. 3, and provides some additional details:

- **Block stacking** requires the robot to pick up the red block and place it on the green block. This is the simplest task, since the robot only has to adapt to new object configurations during test time, but it still requires the robot to precisely localize and grasp the (small) red block.

  We evaluated agents on this task using 25 test positions for the red/green block. These test positions were kept fixed for all policies to ensure maximum reproducibility.

- **Pouring** requires the robot to lift the cup and pour almonds in the target bowl. During test time the cup and target bowls are both novel objects (unseen during training), and are placed in random locations. Thus, this task forces the robot to generalize to new visual inputs.

  We evaluated 3 separate cup/target bowl pairs in 5 positions each (so 15 trials total). Note that none of these objects or positions were seen during test time. Again, the object and position combinations were kept fixed across every model tested.

- **Toasting** is the final task, and it requires the robot to pick up the object, place it in the toaster, and then shut the toaster. During test time, we use a novel object and randomize both the object's initial pose and the toaster's initial orientation. This is the most difficult task, since it requires the robot to execute a multi-stage manipulation strategy, while also generalizing to new visual scenarios.

  We evaluated 2 target objects pairs and randomized the toaster orientation into 5 separate poses (so 10 trials total). Note that none of these objects or toaster orientations were seen during test time. As before, all the test conditions were shared across all policies.

| | Task | **Single Dataset Models (1M Images)** | | | | | **Baselines** |
| | | ImageNet | Ego4D | Kinetics | 100 DoH | RoboNet | R3M [1] |
|---|---|---|---|---|---|---|---|
| **Real** | Block Stacking | $60\% \pm 10\%$ | $52\% \pm 10\%$ | $60\% \pm 10\%$ | $76\% \pm 8.7\%$ | $56\% \pm 10\%$ | $4\% \pm 4\%$ |
| | Pouring | $25\% \pm 11\%$ | $13\% \pm 8.8\%$ | $22\% \pm 11\%$ | $25\% \pm 12\%$ | $6\% \pm 6\%$ | $19\% \pm 10\%$ |
| | Toasting | $10\% \pm 10\%$ | $10\% \pm 10\%$ | $10\% \pm 10\%$ | $30\% \pm 10\%$ | $0\% \pm 0\%$ | $0\% \pm 0\%$ |
| | **Average (Real)** | $\mathbf{32}\% \pm 6.9\%$ | $25\% \pm 6.6\%$ | $\mathbf{31}\% \pm 6.9\%$ | $\mathbf{44}\% \pm 7.1\%$ | $21\% \pm 6.5\%$ | $8\% \pm 3.8\%$ |

Table 4: This table analyzes if Table 1's conclusions apply to different pre-training schemes, or if they are limited to MAE [10]. Specifically, we apply a contrastive visual pre-training algorithm (SimCLR [13]) to 1M images from each of the target datasets. We also add an additional baseline – R3M [1] – that was trained using temporal contrastive learning on Ego4D clips. We evalaute these representations on our 3 real world tasks, and report results as success rates for each task w/ standard error (i.e., Success% ± Std. Err.%). This experiment reveals that the trends do generalize to different pre-training schemes (e.g., vision datasets still stronger than Ego4D), and that the MAE representations are stronger on average.

## D    Replication with SimCLR

Our results from Sec. 4.1 raise questions about several key assumptions in the field. For example, we find that visuo-motor representations learned on the classic ImageNet [2] dataset are stronger than those learned on Ego4D [19] (in-the-wild data) and RoboNet [37] (random robot interactions). But are these trends fundamental to the data, or are they just a quirk of the specific pre-training algorithm/network?

To test this, we repeat the real world evaluation from Table 1 using the SimCLR [63] pre-training algorithm and ResNet-18 architecture [64]. As a refresher, SimCLR is a contrastive learning algorithm that optimizes a network $R$ to "pull together" different views of the same image (i.e., two randomly augmented versions of the same image: $R(z_i), R(z_i^*)$) and "push apart" different images from each other ($R(z_i), R(z_j)$). This is accomplished with the following loss function, where $sim(x, y) = x^T y / (|x||y|)$:

$$L = -log \frac{exp(sim(R(z_i), R(z_i^*))/\tau)}{\sum_{i \neq j} exp(sim(R(z_i), R(z_j))/\tau)}$$

This SimCLR pre-training scheme is applied to each of the 1M images from our target datasets, using the same hyperparameters from the original paper [13].

We compare the newly trained representations alongside an additional ResNet-18 baseline, R3M [1], which was also trained using contrastive learning applied to Ego4D. The results for real world tasks are presented in Table 4. Note that the trends we found in the ViT + MAE evaluations are replicated in these ResNet + SimCLR experiments: **the vision datasets – ImageNet/Kinetics/DoH – create stronger visuo-motor features w/ SimCLR compared to Ego4D/RoboNet!** We also find that despite additional tuning, which was not given to **any** other model (including trying the bigger R3M ResNet-34/50 architectures), the R3M baseline struggles heavily on our tasks (especially stacking). Finally, we note that the average performance of MAEs in Table 1 is stronger than the SimCLR performance ($36\%$ v.s. $31\%$), which further justifies our choice of setup. It is unclear if this is because of the pre-training scheme or architecture choice.

## E    ImageNet Diversity Ablations

One potential hypothesis that would explain our results is that the dataset's *diversity* is critical for effective visuo-motor pre-training. This explanation is intuitive, since information compression is the basis of most self-supervised pre-training algorithms – e.g., MAEs [10] are based upon reconstructing a whole image from an encoding calculated from patches of the image. Thus, a cleaner and more diverse data distribution (like cureated ImageNet dataset) will make pre-text compression task "harder," which in turn could result in a stronger, more robust visuo-motor representation.

While this hypothesize is attractive, the main results in our paper are not able to evaluate its veracity. Thus, we've added an additional experiment to try and shed some light on this theory. Specifically, we take two 500K subsets from ImageNet [2] that have varying levels of diversity. The first, **IN-500K-500C** consists of 500 classes

each with 1000 images (500K frames total). The second, **IN-500K-1000C** uses all 1000 ImageNet classes with 500 images sampled from each (again 500K frames total). Note that these two datasets *are the same size, but the second dataset is more diverse (2x more classes)!* Thus, if diversity is critical, we should expect the 2nd dataset to perform better even though it's the same size.

We evaluate these two models on our real world tasks and present the success rates in Table. 6. Note how the more diverse representation (IN-500K-1000C) performs better on the Pouring and Toasting tasks (w/ equal performance on Stacking), resulting in marginally better performance overall (46% vs 37%). In other words, **keeping all else equal a more diverse pre-training set results in a** 7% **performance boost!** While this result isn't fully definitive, it is an encouraging sign in favor of the diversity hypothesis. However, further work is needed to test this hypothesis in more settings.

| Task | IN-500K-500C | IN-500K-1000C |
|---|---|---|
| Stacking | **70**% | **70**% |
| Pouring | 16% | **32**% |
| Toasting | 25% | **32**% |
| **Average** | 37% | **46**% |

Figure 6: This table compares two representations trained on the same number of frames from ImageNet, but with different diversity levels (500 classes vs 1000). We find that the more diverse image set results in a marginally stronger representation.

