# OpenReview forum: "An Unbiased Look at Datasets for Visuo-Motor Pre-Training"
_robot-learning.org/CoRL/2023/Conference — CoRL 2023 Poster_

### Official Review · Reviewer_35RN · 2023-07-07

**Confidence:** 3
**Originality:** Fair
**Technical Quality:** Fair
**Clarity Of Presentation:** Good
**Impact:** 3

**Recommendation:**

Weak Reject: I recommend rejecting the paper, but will not argue for my recommendation if the majority of other reviewers have a different opinion.

**Review:**

Strengths:
- This is an important problem for real world robot learning.
- The paper does provide a nice suite of experiments in simulation and on a real Franka robot.

Weaknesses:
- The conclusions in the paper are a bit misleading. The evaluation is specific to the representation learning method (MAE) and architecture used. Papers like R3M (Nair et al. 2022) showed that training using time-constrastive learning and other losses significantly outperform vanilla MAE approaches. They also compare pretraining with Ego4D vs ImageNet in their work and find that training with Ego4D is better. The authors should include a much more thorough discussion of these past analyses in prior work in the related works section.
- In order to make the claims it does about ImageNet vs Ego4D, the paper should do a more thorough evaluation using different representation learning losses and architectures, especially since there have been many more new papers on pretraining, and with tricks beyond dropout.
- Would be nice to have more conclusions and analyses on the properties of these datasets that make them more suitable for robot training beyond just "increased visual diversity".

Other:
- Would be good to see standard error on the results.
- Bold important numbers in the tables.

**Quality Of The Limitations Section:**

Limitations are addressed clearly

**Questions For Rebuttal:**

See weaknesses.

**Robotics Focus:**

Sufficient demonstration on hardware

**Summary Of Paper:**

This paper studies the problem of what qualities in datasets are important for effective representation learning for vision-based robot learning systems. Recent representation learning methods have trained on the large Ego4D dataset focusing on human-object interaction, but this paper finds that when used in MAE pretraining, standard image datasets such as ImageNet can perform better than Ego4D even with much fewer frames since the image distribution is diverse. The authors study the effects of using dropout and find that in their evaluations, higher performance in simulation does not always transfer to real world performance.

**Summary Of Recommendation:**

Due to the weaknesses in the evaluation, I think this paper needs more thorough experimentation and analyses before publication at a conference.

---

### Official Review · Reviewer_FvgU · 2023-07-19

**Confidence:** 4
**Originality:** Good
**Technical Quality:** Very Good
**Clarity Of Presentation:** Very Good
**Impact:** 3

**Recommendation:**

Weak Accept: I recommend accepting the paper, but will not argue for my recommendation if the majority of other reviewers have a different opinion.

**Review:**

Originality: The paper provides an unbiased fresh perspective on what datasets to use for pretraining in robotics manipulation, evidence that sim results do not correlate with real results, as well as some insights into improving performance. This is somewhat original in that the findings are somewhat invalidating common wisdom that ego4d is a better choice for pretraining than imagenet for example.

Quality: method is sound. Same SOTA self-supervision MAE is used for all datasets. Then 2-layer policy is trained on top of pretrained encoder. Entire model is finetuned end to end. This is similar to prior work with the addition of using dropout and data augmentation.
I appreciate the investigation into dataset balancing and finding that training on the balanced dataset soup leads to more robustness and performance.

Clarity: paper is well written and very clear. Good amount of details in appendix for reproducibility.

Significance: The number of evaluated tasks remains somewhat small, one can still wonder how general the findings are. It could however inform future data collections efforts to focus on diversity rather than volume for example, which could lead to significantly more general representations and higher performance and robustness.

Relevance: highly relevant to learned robotic manipulation.

**Quality Of The Limitations Section:**

Limitations are addressed clearly

**Questions For Rebuttal:**

- I would like to see standard deviation numbers in table 1 to get a sense of the noise in this evaluation, this would increase confidence in these results. Table 1 should make it more clear that these are success rates, the information is not available unless searching in the text.

- It would be interesting to hear how the authors see the path forward, i.e. focusing on gathering more internet data vs collecting more in-domain data but in a more diverse fashion? Are there any recommendations for the next ego4d-like collection effort?


**Robotics Focus:**

Sufficient demonstration on hardware

**Summary Of Paper:**

The authors seek to verify if the common wisdom of using out-of-domain dataset Ego4D as pretraining of visual stacks for robotics manipulation holds. They compare 5 different datasets using MAE self-supervision pretraining. Then finetuning each pretraining with BC on small in-domain dataset (<= demonstrations).
They find that traditional vision datasets like imagenet yield more general representations than ego4d with less data. The insight is that diversity in image distribution is more important than the size of the dataset. A soup of datasets with 1M examples works as well as with 2M examples. The DoH dataset added to the soup-1M yields a big boost in perf. Authors hypothesize that despite having bigger domain gap with robotics, datasets sampled from the internet (as opposed to actively collected in robotics settings) have more diversity which explains the higher performance.
Authors provide some recipes leading to improved performance, with code and models.


**Summary Of Recommendation:**

A good well-written pape with an interesting meta and unbiased analysis questioning common wisdom around how to do pretraining for robotics. Some good experimental recipes such as insights into how to train dataset soups. It is a welcomed paper with some impact, however it does not seem groundbreaking either.

---

### Official Review · Reviewer_SPNn · 2023-07-20

**Confidence:** 4
**Originality:** Good
**Technical Quality:** Very Good
**Clarity Of Presentation:** Very Good
**Impact:** 3

**Recommendation:**

Weak Accept: I recommend accepting the paper, but will not argue for my recommendation if the majority of other reviewers have a different opinion.

**Review:**

The conclusion of the submitted paper is that

- Standard computer vision datasets, such as ImageNet, surprisingly yield strong performance for robotic pre-training. The simple ImageNet representations outperformed both Ego4D representations and representative baselines in the field. The key insight is that the diversity of the image distribution plays a more significant role than the raw number of images during pre-training.

- Building on the insight above, the authors explored strategies to scale the data distribution by carefully mixing data from different sources. As part of their investigation, they trained a final model called Soup-1M + 1M DoH, which demonstrated a 30% improvement over the baselines on real-world robotic tasks.

- The paper demonstrates how simple regularization techniques like Dropout can enhance real-world performance.

- Results obtained in simulation environments do not necessarily transfer directly to real-world tasks.

Strengths:

- The authors conducted various experiments supporting their claims.

- The paper is clearly presented and well-organized.

Weaknesses:

- There are remaining questions the paper needs to discuss about. Please check below.

**Quality Of The Limitations Section:**

Limitations are addressed clearly

**Questions For Rebuttal:**

- The authors use MAE to pretrain on all the datasets. Will the conclusion hold for other pre-training methods like DINO [A1]? A toy experiment should be good enough.

- When talking about diversity of the datasets, the authors mainly focus on the dataset source. It will be interesting to see the impact of the diversity within a dataset: for example, the performance difference between pre-training on 500 classes of ImageNet v.s. 1000 classes of ImageNet.

- What is the proportion of selected frames among all frames of Ego4D, Kinetics and other video datasets? Though the authors control the total number of samples selected from each dataset, which is a good practice, larger datasets like Ego4D might be harder to learn. Because the larger dataset may contain more objects, however, the number of samples of each object is much lower than ImageNet, which may bias the observations. Experiments mentioned in the 2nd question above can be used to address this concern.

- A followup question is why don't we take advantage of the large dataset rather than using a small subset?

- Is there a chance that all the observations are biased due to different camera FOVs? Maybe the advantage of ImageNet is brought by a FOV which is similar to the real world experiments, while images from Ego4D have more distortion (caused by fish-eye effect). It will be better to show examples of each dataset and the images of the test environments, especially when talking about the domain gap.


[A1] Caron, Mathilde, et al. "Emerging properties in self-supervised vision transformers." Proceedings of the IEEE/CVF international conference on computer vision. 2021.

**Robotics Focus:**

Sufficient demonstration on hardware

**Summary Of Paper:**

This paper investigates the impact of dataset biases and implementation choices on visual pre-training for robotics. The authors conducted experiments using MAE representations trained on 15 different data distributions obtained from 5 commonly used datasets. The representations are fine-tuned to solve manipulation tasks (both simulated and real) via behavior cloning.

**Summary Of Recommendation:**

I would like to give a borderline accept for now, but it will be much better if the authors can address at least 1 or 2 concerns listed above.

---

### Official Review · Reviewer_x3Nd · 2023-07-27

**Confidence:** 4
**Originality:** Very Good
**Technical Quality:** Very Good
**Clarity Of Presentation:** Very Good
**Impact:** 3

**Recommendation:**

Weak Accept: I recommend accepting the paper, but will not argue for my recommendation if the majority of other reviewers have a different opinion.

**Review:**

## Strengths

### Clarity
The paper is easy to follow and well-organized. The contribution is clearly stated. Comprehensive background information (current status of visual pre-training for robotics, how different datasets look like, etc.) are provided. Enough experimental details are presented for reproducing results.

### Quality
The paper is technically sound to me. Claims are supported with experiments. Methods are used appropriately.

### Significance
I think this paper can help the community explore the correct pre-training recipe for various robotics tasks.

## Weaknesses

### Typos
- Figure 2's caption: "from the both the" -> "from both the"

**Quality Of The Limitations Section:**

Limitations are addressed clearly

**Questions For Rebuttal:**

N/A

**Robotics Focus:**

Sufficient demonstration on hardware

**Summary Of Paper:**

This paper benchmarks the effectiveness of visual-pretraining across several well-known datasets. The finding is interesting: pre-training on certain datasets like Ego4D/RoboNet actually can hurt the downstream task performance. Also, blindly increasing the number of frames while not paying attention to the underlying distribution can result in ineffective upscaling.


**Summary Of Recommendation:**

I recommend weak accept as I think the paper presents enough interesting findings for the community.

---

### Author Response · Authors · 2023-08-09
**(August, 9th) General Response**

We thank the reviewers for their detailed comments and suggestions that will strengthen our paper. Also, we're grateful for positive comments from the reviewers such as: “The paper is technically sound,” (x3Nd);  “The paper is clearly presented and well-organized,” (SPNn); “[I] appreciate the investigation into dataset balancing and finding that training on the balanced dataset soup leads to more robustness and performance,” (FvgU); and that we tackle, “an important problem for real world robot learning.” (35RN)

Beyond our detailed individual responses, we include a common response to all reviewers below.

### [New Experiments]: Replicating Results w/ Contrastive Pre-Training and Added R3M Baseline

We present the following new experiments:
1. Reviewers SPNn and 35RN pointed out that our current experiments focus on the MAE pre-training algorithm. To broaden our contributions, we repeat our “dataset-in-a-vacuum” experiments (Table 1) on the real robot, using a *Contrastive Pre-Training Algorithm* (SimCLR [1]) w/ ResNet architecture, to complement our choice of MAE + ViT. This tests if our conclusions can generalize to starkly different pre-training schemes and visual encoders.
2. Reviewer 35RN suggested a comparison with R3M (Nair et al. [2]). Thus, we’ve added R3M as a baseline.

The full results are presented in the table below. The key takeaways are as follows:
* **[Dataset Level Trends Replicate Across Methods/Architectures]:** The vision datasets (ImageNet/Kinetics/100 DoH) still perform better than Ego4D/RoboNet, *even when using a different pre-training scheme/network.* This adds strength to our findings, and indicates that they are not limited to MAE!

* **[SimCLR Representations are Weaker than MAE]:** Representations trained with  SimCLR perform worse on average than representations trained with MAE (31% vs 36%).  We also found them to be *significantly more sensitive to lighting variations and background distractor objects* during evaluation, which made testing very difficult. These observations further validate our choice of MAE in the original experiments. Note, these differences could be a function of the  pre-training algorithm (SimCLR vs MAE) and/or the network architecture (ResNet vs ViT).

* **[R3M Struggles in our Evaluations]:** We find R3M performed significantly worse than the MAE models (50% worse than our Soup-DoH). In particular, it struggled with the stacking task that most other models succeeded in. This is despite specialized tuning efforts on our end, which we didn’t do for *any* other model. One possible explanation is the original paper mostly tested in sim, so its results did not transfer to our real world tasks.

#### [Caption] The table below replicates our paper’s initial experiments (see Table 1) using SimCLR instead of MAE as the pre-training algorithm. Note that the standard vision datasets (bolded) perform better than Ego4D/RoboNet on average! Performance is reported as **Success% \pm Std. Err.** for each task.

|    Model | ImageNet-1M (SimCLR) | Ego4D-1M (SimCLR) | Kinetics-1M (SimCLR) | 100 DoH-1M (SimCLR) | RoboNet-1M (SimCLR) | R3M (Baseline) |
|------------- |-------------------------------|---------------------------------| -------------------------------|---------------------------------| -------------------------------|---------------------------------|
|  Stacking |       60% \pm 10%        |       52% \pm 10%          |       60% \pm 10%        |       76% \pm 8.7%        |       56% \pm 10%        |       4% \pm 4%        |
|  Pouring |       25% \pm 11%        |       13% \pm 8.8%          |       22% \pm 11%        |       25% \pm 12%        |       6% \pm 6%        |       19% \pm 10%        |
|  Toasting |       10% \pm 10%        |       10% \pm 10%          |       10% \pm 10%        |       30% \pm 10%        |       0% \pm 10%        |       0% \pm 10%        |
|  **Average**  |       **32% \pm 6.9%**        |       25% \pm 6.6%          |      **31% \pm 6.9%**        |      **44% \pm 7.1%**        |       21% \pm 6.5%        |       8% \pm 3.8%        |


[1] Chen, Ting, et al. "A simple framework for contrastive learning of visual representations." International conference on machine learning. PMLR, 2020.

[2] Nair, Suraj, et al. "R3M: A universal visual representation for robot manipulation." Conference on Robot Learning, 2022.

---

> ### Author Response · Authors · 2023-08-14
> **Still Open to Discussion!**
>
> First, we'd like to thank the reviewers for their encouraging comments and helpful feedback that has already greatly improved the quality of our results/manuscript.
>
> We're still open to discussion and further feedback, even as the review period comes to a close. Please let us know if you have any lingering comments/concerns!

---

### Author Response · Authors · 2023-08-16
**Concerns Regarding the Review by 35RN**

Dear AC,

Thanks so much for your time and efforts towards the review process. While we gave our best to objectively respond to 35RN, we would like to bring some issues with the review by 35RN to your attention. These include factual errors about fundamentals of representation learning and prior work.

* The review claimed that Nair et al. (R3M) compared with vanilla MAEs, while they did not.
* The reviewer seems to have mixed up the two paradigms of supervised and self-supervised learning (for ImageNet representations). We refresh these fundamentals in the `Comparison to Prior Work` section of our rebuttal.
* Our contributions were mischaracterized to only studying "ImageNet vs Ego4D." We restate our contributions at the very top of our response to 35RN.

We posted our response as early as Aug 9th and despite two attempts, 35RN did not engage in the open discussion phase or even acknowledge reading our response.

We have kept this note intentionally brief. All details are in our response to 35RN, including new experiments that further strengthen our contributions. Thanks again!

---

### Decision · Program_Chairs · 2023-08-30

**Decision:**

Accept (Poster)

**Comment:**

The authors present a study of a wide variety of visual pre-training choices for subsequently training visuomotor policies.  The paper submitted first draft focused on the use of vanilla MAE self-supervision, but swapping out the choice of dataset, then subsequently training a 2-layer policy on top.  In the rebuttal the authors have also added the case of using SimCLR self-supervision instead of MAE self-supervision.  The results show that there is an interesting spread of outcomes form choosing different datasets for pre-training.  For one, they emphasize the importance of the nature of the data distribution, not only its size.  Importantly, as the authors put it well in their rebuttal, they evaluate whether we should "uncritically prefer Ego4D for robot pre-training".

The reviewers had a variety of opinions on the work, but the majority of the reviewers responded positively to the findings, the clear writing, the overall technical soundness, and significance.  Reviewers had several good questions which were engaged with robustly by the authors.  The authors were able to help clarify some aspects of the comparison with the settings studied in some prior work, for example R3M. As the authors confirmed, in R3M the use of ImageNet as pretraining was evaluated in the form of using ImageNet's supervised labels, but the authors here investigate using self-supervised MAE on ImageNet, shown by He at al to produce stronger downstream representations.  They also show that the prior works of R3M and MVP come to opposite conclusions on which pretraining is better -- I don't think we can consider this debate settled at all, but the authors of the present work add additional data, as well as a broader set of data, to this comparison.  They also compare with other useful baselines, like training from scratch, and VC-1.  Overall it is compelling to see how such a simple MAE pretraining option, on a variety of datasets, can perform quite well compared to all of these baselines.  Reviewer 35RN remained suggesting a weak reject, but did agree that the rebuttal updates helped improve the paper.  Their remaining suggestions seem within scope for revising into a final camera-ready submission.

Due to the overall value of the work, I'm recommending acceptance for the paper, with the 4 following required TODOs for the authors:

1. This was not mentioned by any of the reviewers, but I think it is critical. The authors should **remove almost all mentions of the word "diversity"** in the paper, other than as mentioning it as a hypothetical and unevaluated hypothesis. The reason is that diversity is not defined, in any precise sense.  It is true that the nature of the data distribution seems to be having an effect which may be more impactful than purely the dataset size, but it is *not* evaluated whether or not this related to any technical, quantitative definition of diversity.  I would like to point out that Ego4D may actually be "more diverse", in some technical sense, than ImageNet -- I'm not saying it is, but it's not evaluated.  Perhaps really the reason that ImageNet is performing better in pre-training than Ego4D is that ImageNet images have the right level of structure in the image distributions to make the MAE task doable, yet still hard.  If the conditional distribution of pixels modeled in MAE (i.e. p(rest-of-pixels|some-pixels) is too close to a uniform distribution, there is nothing to be gained from MAE pre-training. Ego4D uses ego-motion cameras, which produce a challenging image distribution in which it is hard to predict some pixels from the others -- meanwhile, ImageNet images are from much cleaner "internet image" distributions.  This I think is both incredibly important but easy to fix in about 3 minutes, just ctrl+f for all 'diversity' and adjust for each mention of diversity, for example as follows in the abstract: "the pre-training dataset’s *diversity* matters more than its sheer size" --> "the pre-training dataset’s *distribution* matters more than its sheer size".  Further more insight could be gained by actually analyzing at the MAE loss on the different datasets.
2. The authors should clarify what exactly is the difference between the set of MAEs they are studying, and the "MVP" baseline.  It's my impression that intentionally these are exactly the same, other than "MVP" refers to a specific set of weights which was trained by the authors of that work, on a particular mixture of images.
3. The authors should clarify which images used for the real-robot experiments! In the videos I see there is a wrist-mounted camera, but also a fixed-mount camera, and I don't see where it specifies which (or both) is used.  Critical! Especially since one of these is ego-motion, one is not.
4. The authors should, as reviewer 35RN requests, "provide more qualifying statements throughout the paper, so that it is clearer what specifically is being compared and how that relates to other existing work."